# Discovery of Novel 3,4-Dihydro-2(1*H*)-Quinolinone Sulfonamide Derivatives as New Tubulin Polymerization Inhibitors with Anti-Cancer Activity

**DOI:** 10.3390/molecules27051537

**Published:** 2022-02-24

**Authors:** Juan Ma, Guo-Hua Gong

**Affiliations:** 1Pharmaceutical Engineering College of Guangdong Food and Drug Vocational College, Guangzhou 510000, China; majuan@gdyzyedu.cn; 2Inner Mongolia Key Laboratory of Mongolian Medicine Pharmacology for Cardio-Cerebral Vascular System, Tongliao 028000, China; 3Institute of Mongolia and Western Medicinal Treatment, Affiliated Hospital of Inner Mongolia University for Nationalities, Tongliao 028000, China

**Keywords:** synthesis, quinoline-sulfonamide, anti-tumor, tubulin polymerization

## Abstract

In this paper, a small series of novel quinoline sulfonamide derivatives was synthesized, and their structure of the target compounds were confirmed by 1H NMR and MS. The screening of the news target compounds’ in vitro cytotoxic activities against tumor cell lines by the MTT method was performed. Among them, compound **D13** (N-(4-methoxybenzyl)-2-oxo-N-(3,4,5-trimethoxyphenyl)-1,2,3,4-tetrahydroquinoline-6-sulfonamide exhibited the strongest inhibitory effect on the proliferation of HeLa (IC_50_: 1.34 μM), and this value correlated well with the inhibitory activities of the compound against tubulin polymerization (IC_50_: 6.74 μM). In summary, a new type of quinoline-sulfonamide derivative with tubulin polymerization inhibitory activity was discovered, and it can be used as a lead compound for further modification.

## 1. Introduction

The mortality rate and the incidence of cancer are increasing year by year. The number of patients who die of malignant tumors worldwide has risen to the second place among various causes of death, which seriously threatens human health [1,2]. A microtubule is one of the main components of the cytoskeleton, which exists in almost all eukaryotic cells [3], and has a variety of biological functions, such as participating in intracellular signal transduction [4], material transport and organelle transport [5], chromosome movement and regulating cell division [6]. The biggest difference between cancer and normal cells is that the proliferation of cancer cells is abnormally frequent and often uncontrolled, which makes tumor cell growth heavily dependent on the dynamic instability of tubulins/microtubules involved in polymerization and depolymerization [7,8]. In addition, the rapid proliferation, invasion and metastasis of tumor cells depend greatly on the supply of nutrients in the surrounding blood vessels, so blood vessel proliferation in tumor tissues is obvious. However, the endothelial cells involved in tumor angiogenesis are immature and need a skeletal network composed of microtubules to maintain their morphology [9,10]. Therefore, by inhibiting the polymerization of tubulin into microtubules during tumor cell division, or inhibiting the depolymerization of microtubules into microtubules, mitosis will be unable to proceed or stop, and finally induce the occurrence of apoptosis, so as to achieve the purpose of inhibiting the growth of tumor cells [11]. Since microtubules play a very critical role in the growth and development of tumor cells, they have become an ideal target for anti-tumor drug research [12].

By 2016, there were already seven active binding sites on tubulin. Among them, five binding sites are located on the β subunit of tubulin, including the paclitaxel binding site, laulimalide binding site, vinblastine binding site, maytansine binding site and colchicine binding site. Meanwhile, two binding sites are located on the α subunit of tubulin, including the evipabulin binding site and pironetin binding site [10,13,14,15]. Colchicine can bind to the dimer β subunit in the microtubule lattice and has a strong anti-tumor activity. Importantly, the polyphenol methyl ether structure in its molecule is one of the main pharmacophores that inhibits tubulin polymerization [11,16]. Combretastatin A-4 (CA-4) (Figure 1) is a natural and powerful small molecule anti-mitotic agent and vascular blocker, which is isolated from the bark of South African shrub willow [17,18]. BPROL075 (Figure 1) is an indole tubulin inhibitor and entered preclinical research as an antimitotic agent. It can strongly inhibit tubulin polymerization and has nanomolar inhibitory activity against a variety of tumor cell lines, including drug-resistant ones [19]. Interestingly, the tubulin inhibitor CA-4 and BPROL075 (Figure 1) also contain this structure [20]. In addition, the sulfonamide tubulin inhibitor ABT-751 (Figure 1), which also acts on the colchicine site, has strong proliferation inhibitory activity [19]. Moreover, there are many small molecular compounds with diverse structures, such as quinolinones that target the binding site of colchicine, which show excellent anti-tumor activity and block tumor vasculature in vivo and in vitro, showing good application prospects [21]. These interesting studies have stimulated our interest, so we envisioned putting these effective fragments together, hoping to obtain a series of tubulin inhibitors (Figure 2). Furthermore, the molecular docking and tubulin inhibitory activity of the most promising compound (**D13**) were investigated.

## 2. Results and Discussion

### 2.1. Chemistry

The synthetic route for the targets is shown in Figure 1. Both 3,4-dihydroquinolin-2(1*H*)-one and sulfonic chloride were chlorosulfonated to obtain intermediate **A**. Moreover, 3,4,5-trimethoxyaniline and different substituted aldehydes were subjected to the Schiff’s base reaction to obtain different intermediates **B** [21,22,23]. Subsequently, intermediate **B** was reduced to obtain **C**. The target compounds **D1–D16** were obtained by the nucleophilic substitution reaction between **A** and **C** [24]. Finally, all the target compounds were fully characterized by NMR and MS which was showed in the Appendix A.

### 2.2. Biological Evaluation

#### 2.2.1. In Vitro Anticancer Activity

The MTT method was used to evaluate the anti-tumor activity and cytotoxic activity of all the synthesized compounds in four tumor cell lines (HeLa, HCT-116, A549 and HepG-2) and normal liver cells (L02). At the same time, the 5-fluorouracil(5-Fu) and CA-4 were used as positive references. As shown in Table 1, most of them showed effective inhibitory activity, and IC_50_ < 10 μM for these four cell lines. In particular, the compound **D13** showed the strongest anti-proliferative activity and was better than the positive control 5-fluorouracil. Among them, the IC_50_ for HeLa cells was 1.46 μM, the IC_50_ for A549 cells was 1.46 μM, the IC_50_ for HCT116 cells was 0.94 μM, the IC_50_ value for HCT116 cells was 0.94 μM and the IC_50_ value for HepG-2 cells was 1.82 μM. Unfortunately, our target compound has a similar cytotoxicity to tumor cells and normal cells.

Based on the structure–activity relationship (SAR) study, we tried to prove how the substituents at different positions of the benzene ring affect its anti-cancer activity. As shown in, at the beginning, we first synthesized the ortho and meta substituted compounds, **D7**, **D8**, **D9**, **D11** and **D15**. Unfortunately, they all showed poor antiproliferative activity. Then, we decided to introduce para-substituted benzaldehyde. What is exciting was that all the compounds (**D1**, **D2**, **D3**, **D4**, **D5**, **D6** and **D13**) showed strong anti-proliferative activity. More importantly, in Hela and HepG-2 cells, the anti-proliferative activity of all the compounds was better than the positive control 5-Fu. Additionally, in the four cell lines, the order of the types and positions of the substituents on the benzene ring to enhance the antiproliferative activity was 4-OCH_3_ > 4-CH_3_ > 4-N(CH_3_)_2_ > 4-Br > 4-Cl > 4-F > 4-H; from this we drew the clear conclusion that the introduction of electron-donating groups at the para position of the benzene ring was more active than the electron-donating groups. In addition, the compound **D13** with the 4-OCH_3_ substitution showed the strongest anti-proliferative activity. This stimulated our interest, so we continued to introduce -OCH_3_ substituted at different positions to explore the effect of -OCH_3_ substituted at different positions on the anti-tumor activity of the compound. Unfortunately, all the compounds (**D11**, **D12**, **D14** and **D15**) except for **D16**, exhibited poor antiproliferative activity. In short, we obtained a compound named **D13** with good anti-tumor activity, and it was worthy of further study.

#### 2.2.2. Inhibition of Tubulin Polymerization of **D13**

Compound **D13** showed the strongest anti-proliferative activity. Therefore, we used the tubulin polymerization detection kit to further verify its inhibitory effect on tubulin polymerization. CA-4 was used as the reference compound; in addition, the control group was also set. As shown in Figure 3, the compound **D13** strongly inhibited the tubulin assembly assay with an IC_50_ of 6.74 μM. These results suggest that the compounds may inhibit the cell growth through tubulin polymerization inhibition. Unfortunately, it did not surpass CA-4 (2.64 μM).

### 2.3. Docking Analysis

We used the computer-aided drug design software Discovery Studio 2017 Server for the molecular model construction and protein structure treatment to complete the docking of the target compounds with the tubulin (Tubulin-ABT751, 3HKC. pdb). In order to study whether the binding ability of the target compound to the receptor was positively correlated with its anti-proliferative activity, we selected the compound **D13** with the best anti-proliferative activity, **D5** with less activity and **D15** with the worst activity. The results are shown in Figure 4, where the compound **D13** has the strongest binding ability to the tubulin receptor and shows the lowest binding energy (CDOCKER_INTERACTION_ENERGY = −53.11 kJ/mol). At the same time, the binding ability of compound **D5** and compound **D15** decreased successively to −37.41 kJ/mol and −31.5 kJ/mol, respectively. This result indirectly verifies that the target compound inhibits the proliferation of tumor cells by inhibiting the polymerization of tubulin.

## 3. Conclusions

With the aim to further explore the novel tubulin polymerization inhibitor, a small series of novel quinoline sulfonamide derivatives was synthesized, and their antiproliferative activities with their mechanism of action were investigated. In particular, the most potent compound, **D13**, exhibited the best in vitro cytotoxic activity in cellular assays with a mean IC_50_ value of 1.34 μM on the HeLa human tumor cell line, and significant potency against tubulin assembly with an IC_50_ value of 6.74 μM. Moreover, the results of the molecular docking study showed that compound **D13** had a strong binding affinity to tubulin and may have played a crucial role in inhibiting its activity. In summary, a new type of quinoline-sulfonamide derivative with tubulin polymerization inhibitory activity was discovered, and the most active compound, **D13**, can be used as a lead compound for further modification.

## 4. Experimental Section

### 4.1. Chemistry General Methods

The raw materials used in the experiment were purchased from Aladdin reagent. The progress of the reaction was monitored by thin-layer chromatography, and a chromatographic column was used for separation. The melting point of the target product was measured in an open capillary tube (the temperature is not corrected), ^1^H-NMR was used the chemical shift of TMS as the zero point, measured by AV-300 nuclear magnetic resonance instrument, and the mass spectrum was measured with a flight mass spectrometer.

### 4.2. Procedure for the Synthesis of Compound ***B***

3,4-dihydroquinolin-2(1*H*)-one (10 g, 68.03 mmol) was added to a 50 mL found-bottom flask and then stirred with 9 mL of sulfonic chloride (as the solvent) at 80 °C for 6–8 h, and the reaction was monitored by TLC. After the reaction was over, we poured it into ice water and a solid precipitated out to obtain compound A. White powder; yield: 86.5%; m.p. 209–212 °C. ^1^H NMR (300 MHz, DMSO-*d_6_*,ppm) *δ* 10.15 (s, 1H, -CONH-); 7.44 (d, 1H, *J* = 1.5 Hz, Ar-H); 7.41 (d, 1H, *J* = 9.0 Hz, Ar-H); 6.84 (d, 1H, *J* = 9.0 Hz, Ar-H) and 2.42–3.02 (m, 4H, -CH_2_CH_2_-).

### 4.3. General Synthesis Process of Compounds ***C1**–**C16***

3,4,5-trimethoxyaniline (500 mg, 2.73 mmol) and different substituted benzaldehydes (3.28 mmol) were added to a 25 mL found-bottom flask and then stirred with 5 mL of methanol (as the solvent) at 50 °C for 6–8 h. After the end of the reaction was detected by TLC, we added NaBH_4_ (4.00 mmol) to the reaction solution and continued to react for 4 h. The mixture was extracted 3 times with 15 mL of ethyl acetate and an appropriate amount of saturated brine. The organic phase was concentrated under reduced pressure and purified by silica gel chromatography to obtain compounds **C1**–**C16**. White powder; yield 65–83%. The ^1^H NMR of compound **C1**: (300 MHz, CDCl_3_, ppm) *δ*7.40–7.38 (m, 3H, Ar-H), 7.35–7.28 (m, 2H, Ar-H), 5.90 (s, 2H, Ar-H), 4.32 (s, 1H, -NH), 3.81 (s, 6H, -COH_3_) and 3.78 (s, 3H, -COH_3_).

### 4.4. General Synthesis Process of Compounds ***D1**–**D16***

Compound **C** (1 mmol) and compound **B** (1.1 mol) were added to a 25 mL found-bottom flask and then stirred with 5 mL of acetonitrile (as the solvent) at 80 °C for 8 h; the mixture was extracted with ethyl acetate. The organic phase was concentrated under reduced pressure and purified by silica gel chromatography to obtain compounds **D1**–**D16**.

#### 4.4.1. N-benzyl-2-oxo-N-(3,4,5-trimethoxyphenyl)-1,2,3,4-tetrahydroquinoline-7-sulfonamide (**D1**)

White powder; yield 52%; m.p. 96–98 °C. ^1^H NMR (300 MHz, CDCl_3_, ppm) *δ* 8.36 (s, 1H), 7.60–7.56 (m, 2H), 7.27 (s, 5H), 6.88 (d, *J* = 8.2 Hz, 1H), 6.17 (s, 2H), 4.70 (s, 2H), 3.81 (s, 3H), 3.66 (s, 6H), 3.03 (t, *J* = 7.5 Hz, 2H) and 2.71 (t, *J* = 6.7 Hz, 2H). ^13^C NMR (75 MHz, CDCl_3_, ppm) *δ* 170.88, 152.74 (2C), 142.30, 137.56, 135.96, 134.54, 131.75, 128.49 (2C), 128.25 (2C), 127.59, 127.41, 123.63, 115.46, 106.53 (2C), 60.65, 55.95 (2C), 55.07, 30.23 and 25.10. MS (*m/z*) calculated for C_25_H_27_N_2_O_6_S^+^[M+H]^+^: 483.16, found: 483.18.

#### 4.4.2. N-(4-bromobenzyl)-2-oxo-N-(3,4,5-trimethoxyphenyl)-1,2,3,4-tetrahydroquinoline-7-sulfonamide (**D2**)

White powder; yield 63%; m.p. 112–114 °C. ^1^H NMR (300 MHz, CDCl_3_, ppm) *δ* 8.64 (s, 1H), 7.57–7.53 (m, 2H), 7.41 (d, *J* = 8.3 Hz, 2H), 7.15 (d, *J* = 8.3 Hz, 2H), 6.90 (d, *J* = 8.2 Hz, 1H), 6.17 (s, 2H), 4.64 (s, 2H), 3.82 (s, 3H), 3.68 (s, 6H), 3.03 (t, *J* = 7.5 Hz, 2H) and 2.71 (t, *J* = 7.6 Hz, 2H). ^13^C NMR (75 MHz, CDCl_3,_ ppm) δ 170.98, 152.92 (2C), 142.32, 137.79, 135.17, 134.41, 131.61, 131.44 (2C), 130.25 (2C), 127.55 (2C), 123.71, 121.60, 115.51, 106.47 (2C), 60.75, 56.06 (2C), 54.49, 30.25 and 25.14. MS (*m/z*) calculated for C_25_H_27_BrN2O_6_S^+^[M+H]^+^: 561.07, found: 561.20.

#### 4.4.3. N-(4-fluorobenzyl)-2-oxo-N-(3,4,5-trimethoxyphenyl)-1,2,3,4-tetrahydroquinoline-6-sulfonamide (**D3**)

White powder; yield 51%; m.p. 108–110 °C. ^1^H NMR (300 MHz, CDCl_3_, ppm) δ 8.56 (s, 1H), 7.56 (d, *J* = 10.5 Hz, 2H), 7.23 (dd, *J* = 8.5, 5.4 Hz, 2H), 7.02–6.87 (m, 3H), 6.15 (s, 2H), 4.66 (s, 2H), 3.82 (s, 3H), 3.67 (s, 6H), 3.03 (t, *J* = 7.6 Hz, 2H) and 2.71 (t, *J* = 7.6 Hz, 2H). ^13^C NMR (126 MHz, DMSO, ppm) δ 170.81, 160.93, 152.91 (2C), 143.11, 137.46, 134.98, 133.22, 130.98, 130.76, 130.70, 127.86, 127.77, 124.66, 115.63, 115.53, 115.46, 106.99 (2C), 60.47, 56.37 (2C), 53.55, 30.34 and 24.85. MS (*m/z*) calculated for C_25_H_27_FN_2_O_6_S^+^[M+H]^+^: 501.15, found: 501.19.

#### 4.4.4. N-(4-chlorobenzyl)-2-oxo-N-(3,4,5-trimethoxyphenyl)-1,2,3,4-tetrahydroquinoline-6-sulfonamide (**D4**)

White powder; yield 61%; m.p. 119–121 °C. ^1^H NMR (300 MHz, CDCl_3_, ppm) δ 8.64 (s, 1H), 7.58–7.53 (m, 2H), 7.28–7.17 (m, 4H), 6.90 (d, *J* = 8.2 Hz, 1H), 6.17 (s, 2H), 4.66 (s, 2H), 3.82 (s, 3H), 3.67 (s, 6H), 3.03 (t, *J* = 7.5 Hz, 2H) and 2.71 (t, *J* = 7.6 Hz, 2H). ^13^C NMR (126 MHz, DMSO, ppm) δ 170.80, 152.92 (2C), 143.14, 137.47, 136.15, 135.00, 132.44, 130.85 (2C), 130.52 (2C), 128.76, 127.88, 127.78, 124.66, 115.53, 106.94 (2C), 60.48, 56.40 (2C), 53.56, 30.34 and 24.84. MS (*m/z*) calculated for C_25_H_27_ClN_2_O_6_S^+^[M+H]^+^: 517.12, found: 517.16.

#### 4.4.5. N-(4-methylbenzyl)-2-oxo-N-(3,4,5-trimethoxyphenyl)-1,2,3,4-tetrahydroquinoline-6-sulfonamide (**D5**)

White powder; yield 65%; m.p. 128–130 °C. ^1^H NMR (300 MHz, CDCl_3_, ppm) δ 8.91 (s, 1H), 7.56 (dd, *J* = 12.5, 4.2 Hz, 2H), 7.10 (dd, *J* = 18.2, 8.1 Hz, 4H), 6.97 (d, *J* = 8.3 Hz, 1H), 6.17 (s, 2H), 4.64 (s, 2H), 3.81 (s, 3H), 3.66 (s, 6H), 3.01 (t, *J* = 7.5 Hz, 2H), 2.69 (t, *J* = 7.6 Hz, 2H) and 2.31 (s, 3H). ^13^C NMR (126 MHz, DMSO, ppm) δ 170.82, 153.01 (2C), 143.60, 137.72, 137.23, 134.94, 134.35, 130.75, 130.53, 130.27, 128.02, 127.95, 127.86, 126.00, 124.61, 115.51, 107.63 (2C), 60.70, 56.83 (2C), 56.21, 31.76, 24.87 and 19.22. MS (*m/z*) calculated for C_26_H_29_N_2_O_6_S^+^[M+H]^+^: 497.17, found: 497.17.

#### 4.4.6. N-(4-(dimethylamino)benzyl)-2-oxo-N-(3,4,5-trimethoxyphenyl)-1,2,3,4-tetrahydroquinoline-6-sulfonamide (**D6**)

White powder; yield 63%; m.p. 127–129 °C. ^1^H NMR (300 MHz, CDCl_3_) δ 8.20 (s, 1H), 7.64–7.51 (m, 2H), 7.07 (d, *J* = 8.5 Hz, 2H), 6.86 (d, *J* = 8.3 Hz, 1H), 6.60 (d, *J* = 8.7 Hz, 2H), 6.17 (s, 2H), 4.60 (s, 2H), 3.82 (s, 3H), 3.66 (s, 6H), 3.02 (t, *J* = 7.4 Hz, 2H), 2.92 (s, 6H) and 2.76–2.65 (m, 2H). ^13^C NMR (126 MHz, DMSO, ppm) δ 170.82, 152.79 (2C), 150.20, 142.93, 137.28, 135.13, 131.31, 129.75, 127.79, 127.71, 124.60, 123.74, 115.45, 112.48, 107.01, 60.47, 56.35, 53.93, 30.35 and 24.85. MS (*m/z*) calculated for C_27_H_32_N_3_O_6_S^+^[M+H]^+^: 526.20, found: 526.22.

#### 4.4.7. N-(3-methylbenzyl)-2-oxo-N-(3,4,5-trimethoxyphenyl)-1,2,3,4-tetrahydroquinoline-6-sulfonamide (**D7**)

White powder; yield 67%; m.p. 120–122 °C. 1H NMR (300 MHz, CDCl_3_, ppm) δ 9.14 (s, 1H), 7.57 (dd, *J* = 10.9, 2.6 Hz, 2H), 7.10 (dt, *J* = 19.5, 7.9 Hz, 4H), 6.94 (d, *J* = 8.2 Hz, 1H), 6.18 (s, 2H), 4.66 (s, 2H), 3.81 (s, 3H), 3.66 (s, 6H), 3.03 (t, *J* = 7.5 Hz, 2H), 2.71 (t, *J* = 7.6 Hz, 2H) and 2.30 (s, 3H). ^13^C NMR (126 MHz, DMSO, ppm) δ 170.82, 152.79 (2C), 143.08, 137.48, 137.23, 134.94, 134.35, 130.75, 130.53, 130.27, 128.02, 127.95, 127.86, 126.00, 124.61, 115.51, 107.00 (2C), 60.50, 56.36, 52.64, 45.91, 30.36, 24.87 and 19.22. MS (*m/z*) calculated for C_26_H_29_N_2_O_6_S^+^[M+H]^+^: 497.17, found: 497.18.

#### 4.4.8. N-(2-fluorobenzyl)-2-oxo-N-(3,4,5-trimethoxyphenyl)-1,2,3,4-tetrahydroquinoline-6-sulfonamide (**D8**)

White powder; yield 56%; m.p. 138–140 °C. ^1^H NMR (300 MHz, CDCl_3_, ppm) δ 8.90 (s, 1H), 7.56 (d, *J* = 9.2 Hz, 2H), 7.27–7.19 (m, 1H), 7.12–6.86 (m, 4H), 6.19 (s, 2H), 4.69 (s, 2H), 3.82 (s, 3H), 3.68 (s, 6H), 3.03 (t, *J* = 7.5 Hz, 2H) and 2.71 (t, *J* = 7.5 Hz, 2H). ^13^C NMR (75 MHz, CDCl_3_, ppm) δ 170.88, 16.85, 152.74 (2C), 142.30, 137.56, 135.96, 134.54, 131.75, 128.49 (2C), 128.25 (2C), 127.59, 127.41, 123.63, 115.46, 106.53 (2C), 60.65, 55.95 (2C), 55.07, 30.23 and 25.10. MS (*m/z*) calculated for C_25_H_26_FN_2_O_6_S^+^[M+H]^+^: 501.15, found: 501.18.

#### 4.4.9. N-(2-chlorobenzyl)-2-oxo-N-(3,4,5-trimethoxyphenyl)-1,2,3,4-tetrahydroquinoline-6-sulfonamide (**D9**)

White powder; yield 54%; m.p. 117–119 °C. ^1^H NMR (300 MHz, CDCl_3_, ppm) δ 8.64 (s, 1H), 7.58–7.53 (m, 2H), 7.28–7.17 (m, 4H), 6.90 (d, *J* = 8.2 Hz, 1H), 6.17 (s, 2H), 4.66 (s, 2H), 3.82 (s, 3H), 3.67 (s, 6H), 3.03 (t, *J* = 7.5 Hz, 2H) and 2.71 (t, *J* = 7.6 Hz, 2H). ^13^C NMR (126 MHz, DMSO, ppm) δ 170.83, 152.90 (2C), 143.16, 137.59, 135.05, 134.19, 133.21, 131.50, 130.73, 129.81, 129.76, 127.96, 127.85, 127.62, 124.66, 115.51, 107.00 (2C), 60.51, 56.38 (2C), 52.21, 30.33 and 24.84. MS (*m/z*) calculated for C_25_H_26_ClN_2_O_6_S^+^[M+H]^+^: 517.12, found: 517.15.

#### 4.4.10. N-(3,4-dichlorobenzyl)-2-oxo-N-(3,4,5-trimethoxyphenyl)-1,2,3,4-tetrahydroquinoline-6-sulfonamide (**D10**)

White powder; yield 64%; m.p. 123–125 °C. ^1^H NMR (300 MHz, CDCl_3_, ppm) δ 8.57 (s, 1H), 7.54 (d, *J* = 11.2 Hz, 2H), 7.37 (d, *J* = 7.7 Hz, 2H), 7.14 (d, *J* = 8.0 Hz, 1H), 6.90 (d, *J* = 8.1 Hz, 1H), 6.19 (s, 2H), 4.64 (s, 2H), 3.83 (s, 3H), 3.70 (s, 6H), 3.03 (t, *J* = 7.4 Hz, 2H) and 2.71 (t, *J* = 7.6 Hz, 2H). ^13^C NMR (126 MHz, DMSO, ppm) δ 170.81, 152.99 (2C), 143.22, 138.46, 137.55, 134.95, 131.31, 131.04, 130.63, 130.53, 130.43, 128.94, 127.95, 127.82, 124.69, 115.54, 106.92 (2C), 60.49, 56.43 (2C), 53.07, 30.32 and 24.84. MS (*m/z*) calculated for C_25_H_25_Cl_2_N_2_O_6_S^+^[M+H]^+^: 551.08, found: 551.09.

#### 4.4.11. N-(5-chloro-2-fluorobenzyl)-2-oxo-N-(3,4,5-trimethoxyphenyl)-1,2,3,4-tetrahydroquinoline-8-sulfonamide (**D11**)

White powder; yield 67%; m.p. 124–126 °C. ^1^H NMR (300 MHz, CDCl_3_) δ 8.86 (s, 1H), 7.69–7.54 (m, 2H), 7.14 (dt, *J* = 12.5, 7.9 Hz, 2H), 6.99–6.79 (m, 2H), 6.21 (s, 2H), 4.89 (s, 2H), 3.80 (s, 3H), 3.66 (s, 6H), 3.04 (t, *J* = 7.5 Hz, 2H) and 2.71 (t, *J* = 7.5 Hz, 2H). ^13^C NMR (126 MHz, DMSO, ppm) δ 170.83, 160.87, 152.67 (2C), 143.21, 137.78, 134.58, 130.43, 128.04, 127.95, 126.03, 124.63, 121.95, 121.81, 115.46, 114.96, 114.78, 107.01 (2C), 60.55, 56.22 (2C), 46.42, 30.34 and 24.85. MS (*m/z*) calculated for C_25_H_25_ClFN_2_O_6_S^+^[M+H]^+^: 535.11, found: 535.14.

#### 4.4.12. N-(3-methoxybenzyl)-2-oxo-N-(3,4,5-trimethoxyphenyl)-1,2,3,4-tetrahydroquinoline-6-sulfonamide (**D12**)

White powder; yield 45%; m.p. 121–123 °C. ^1^H NMR (300 MHz, CDCl_3_, ppm) δ 8.63 (s, 1H), 7.59–7.52 (m, 2H), 7.18 (t, *J* = 8.1 Hz, 1H), 6.92 (d, *J* = 8.2 Hz, 1H), 6.84–6.77 (m, 3H), 6.20 (s, 2H), 4.67 (s, 2H), 3.82 (s, 3H), 3.77 (s, 3H), 3.67 (s, 6H), 3.03 (t, *J* = 7.5 Hz, 2H) and 2.70 (t, *J* = 9.0 Hz, 2H). ^13^C NMR (126 MHz, DMSO, ppm) δ 170.80, 159.61, 152.87 (2C), 143.09, 138.59, 137.40, 135.15, 131.00, 129.83, 127.86, 127.76, 124.65, 120.83, 115.52, 114.09, 113.34, 106.93 (2C), 60.47, 56.38, 55.42, 54.14, 45.86, 30.34 and 24.85. MS (*m/z*) calculated for C_26_H_29_N_2_O_7_S^+^[M+H]^+^: 513.17, found: 513.19.

#### 4.4.13. N-(4-methoxybenzyl)-2-oxo-N-(3,4,5-trimethoxyphenyl)-1,2,3,4-tetrahydroquinoline-6-sulfonamide (**D13**)

White powder; yield 48%; m.p. 122–124 °C. ^1^H NMR (300 MHz, CDCl_3_, ppm) δ 8.36 (s, 1H), 7.59–7.53 (m, 2H), 7.15 (d, *J* = 8.6 Hz, 2H), 6.88 (d, *J* = 8.2 Hz, 1H), 6.80 (d, *J* = 8.7 Hz, 2H), 6.16 (s, 2H), 4.63 (s, 2H), 3.82 (s, 3H), 3.78 (s, 3H), 3.66 (s, 6H), 3.03 (t, *J* = 7.5 Hz, 2H) and 2.70 (t, *J* = 9.0 Hz, 2H). ^13^C NMR (126 MHz, DMSO, ppm) δ 170.80, 159.61, 152.87 (2C), 143.09, 138.59, 137.40, 135.15, 131.00, 129.83, 127.86, 127.76, 124.65, 120.83, 115.52, 114.09, 113.34, 106.93 (2C), 60.47, 56.38, 55.42, 54.14, 45.86, 30.34 and 24.85. MS (*m/z*) calculated for C_26_H_29_N_2_O_7_S^+^[M+H]^+^: 513.17, found: 513.19.

#### 4.4.14. N-(3,4-dimethoxybenzyl)-2-oxo-N-(3,4,5-trimethoxyphenyl)-1,2,3,4-tetrahydroquinoline-6-sulfonamide (**D14**)

White powder; yield 49%; m.p. 116–118 °C. ^1^H NMR (300 MHz, CDCl_3_, ppm) δ 8.24 (s, 1H), 7.62–7.52 (m, 2H), 6.89 (d, *J* = 8.1 Hz, 1H), 6.86 (d, *J* = 1.8 Hz, 1H), 6.75–6.65 (m, 2H), 6.17 (s, 2H), 4.63 (s, 2H), 3.85 (s, 3H), 3.84 (s, 3H), 3.82 (s, 3H), 3.67 (s, 6H),3.03 (t, *J* = 7.5 Hz, 2H) and 2.75–2.66 (m, 2H). ^13^C NMR (126 MHz, DMSO, ppm) δ 170.81, 152.84 (2C), 148.92, 148.54, 143.04, 137.36, 135.08, 131.08, 128.95, 127.87, 127.76, 124.62, 121.18, 115.52, 112.28, 111.86, 107.03 (2C), 60.47, 56.38 (2C), 55.82, 53.97, 45.88, 30.34 and 24.85. MS (*m/z*) calculated for C_27_H_31_N_2_O_8_S^+^[M+H]^+^: 543.18, found: 543.15.

#### 4.4.15. O-(2,5-dimethoxybenzyl)-2-oxo-N-(3,4,5-trimethoxyphenyl)-1,2,3,4-tetrahydroquinoline-6-sulfonamide (**D15**)

White powder; yield 62%; m.p. 127–129 °C. ^1^H NMR (300 MHz, CDCl_3_, ppm) δ 8.42 (s, 1H), 7.62–7.50 (m, 2H), 7.04 (d, *J* = 2.6 Hz, 1H), 6.88 (d, *J* = 8.3 Hz, 1H), 6.76–6.67 (m, 2H), 6.30 (s, 2H), 4.74 (s, 2H), 3.81 (s, 3H), 3.76 (s, 3H), 3.69 (s, 6H), 3.63 (s, 3H), 3.03 (t, *J* = 7.6 Hz, 2H) and 2.70 (t, *J* = 7.5 Hz, 2H). ^13^C NMR (126 MHz, DMSO, ppm) δ 170.80, 153.36, 152.84 (2C), 151.47, 143.07, 137.40, 135.55 (2C), 131.20, 127.84, 127.73, 125.70, 124.60, 115.80, 115.49, 113.65, 112.34, 106.89 (2C), 60.50, 56.36, 56.26, 55.81, 49.42, 30.34 and 24.85. MS (*m/z*) calculated for C_27_H_31_N_2_O_8_S^+^ [M+H]^+^: 543.18, found: 543.13.

#### 4.4.16. 2-oxo-N-(3,4,5-trimethoxybenzyl)-N-(3,4,5-trimethoxyphenyl)-1,2,3,4-tetrahydroquinoline-6-sulfonamide (**D16**)

White powder; yield 62%; m.p. 131–133 °C. ^1^H NMR (300 MHz, DMSO, ppm) δ 10.54 (s, 1H), 7.51 (d, *J* = 6.1 Hz, 2H), 7.03 (d, *J* = 9.0 Hz, 1H), 6.54 (s, 2H), 6.32 (s, 2H), 4.65 (s, 2H), 3.68 (s, 6H), 3.56 (dd, *J* = 26.9, 22.2 Hz, 12H), 3.02–2.90 (m, 2H) and 2.50 (s, 2H). MS (*m/z*) calculated for C_28_H_33_N_2_O_9_S^+^[M+H]^+^: 573.19, found: 573.13.

### 4.5. In Vitro Anticancer Experiment

The HeLa, HCT-116, A549 and HepG-2 cells in the logarithmic growth phase were trypsinized, diluted with 10% DMEM medium and evenly seeded in a 96-well plate with 1 × 10^4^ cells per well. We placed the inoculated 96-well plate in an incubator at 37 °C and 5% CO_2_ for 4 h. After the cells adhered to the wall, we removed the medium. We added 150 μL of medicated medium (1% DMEM) to each well of the experimental group, and the control group was added with an equal volume of solvent, with 3 replicate holes for each concentration. Then, we put it in the incubator and continued to incubate for 48 h. After that, we added MTT reagent in the dark, continued incubating for 4 h, discarded the supernatant, added 150 µL DMSO to each well and shook it for 10 min in the dark to fully dissolved the formazan. The absorbance was read at 492 nm by a microplate reader (ELx 800, BioTek, Highland Park, Winooski, VT, USA). 

The inhibition rate (%) = (1-dosing hole OD value/control hole OD value); we drew a logarithmic curve diagram according to the inhibition rate and the dosing concentration to obtain the half inhibitory concentration.

### 4.6. In Vitro Tubulin Polymerization Assay

A tubulin polymerization assay was performed by measuring the increase in the fluorescence intensity, which can be easily recorded due to the incorporation of a fluorescent reporter, DAPI (4′,6-diamidino-2-phenylindole), a fluorophore that is known to be a DNA intercalator. In our experiment, a commercial kit (cytoskeleton, cat. #BK011P) purchased from Cytoskeleton (Danvers, MA, USA) was used for the tubulin polymerization. The final buffer used for tubulin polymerization contained 80.0 mM of piperazine-N,N’-bis(2ethanesulfonic acid) sequisodium salt (pH 6.9), 2.0 mM MgCl_2_, 0.5 mM EGTA, 1.0 mM GTP, and 10.2% of glycerol. First, 5 μL of the tested compounds at the indicated concentrations was added, and the mixture was warmed to 37 °C for 1 min; then, the reaction was initiated by the addition of 55 uL of the tubulin solution. The fluorescence intensity enhancement was recorded every 30 s for 40 min in a multifunction microplate reader (Molecular Devices, Flex Station 3) (emission wavelength of 420 nm, excitation wavelength of 360 nm). The area under the curve was used to determine the concentration that inhibited the tubulin polymerization by 50% (IC_50_), and was calculated using GraphPad Prism Software version5.02 (GraphPad Inc., La Jolla, CA, USA).

### 4.7. Molecular Modeling

Molecular docking was performed using the Discovery Studio (DS) 2017 Software. The protein and ligand samples were prepared, water molecules were deleted and a DS Server added hydrogen (https://www.rcsb.org/structure/3HKC, accessed on 15 October 2021). The docking process was performed according to the CDOCKER protocol, where the technical parameter Pose Cluster Radius was reset to 0.5 and the other parameters were unchanged. The docking of the active site was set to the coordinates x = 39.42, y = 52.17 and z = −9.19 as the center, with a radius of 7.43 Å spheres. The docking result was treated with DS Client.

## Data Availability

All data generated or analyzed during this study are included in this article.

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
