# Peer review of "Discovery of Novel 3,4-Dihydro-2(1H)-Quinolinone Sulfonamide Derivatives as New Tubulin Polymerization Inhibitors with Anti-Cancer Activity"

_molecules, 2022, doi:10.3390/molecules27051537_

Round 1

Reviewer 1 Report

In this submitted manuscript titled “Discovery of Novel Quinoline-sulfonamide Derivatives as New Tubulin Polymerization Inhibitors with Anti-Cancer Activity”, Ma et al reported the synthesis of a series of 3,4-Dihydro-2(1H)-quinolinone sulfonamide derivatives and evaluated their anti-proliferative activity in four tumor cell lines. The authors found one of the derivatives called D13 showing the most potent activity and having IC50s at low-uM range. The authors also tried proving these derivatives are targeting microtubule dynamics and therefore performed tubulin polymerization assay and molecular modeling with the best compound D13.

The chemical structures and their anti-proliferative activities in this manuscript are of great interest. It is nevertheless both the polymerization assay and molecular modeling are not satisfying. At this stage, I do not recommend its publication in Molecules and revised version is required:

Major points:

  • The authors employed 5-fluorouracil as a positive control in Table 1, it is nevertheless 5-fluorouracil has an unrelated pharmacological target to the synthesized compounds presented in this manuscript. Therefore, I think it is inappropriate and I would suggest the authors to switch to tubulin inhibitors as positive controls.
  • The tubulin polymerization result is not convincing to me that D13 is inhibiting tubulin polymerization, also the positive control CA-4 does not display apparent tubulin polymerization activity as it supposed to. Can the authors please redo this assay or switch to assay measuring O.D. values?
  • Molecular modeling: It is impossible for the proton of OCH3 to form hydrogen-bonding interaction, please redo the modeling with the help from expert.

Minor points:

  • In line 48, the authors state “Four tubulin inhibitor binding sites have been identified on microtubules, including paclitaxel binding site, Laulimalide binding site, vinblastine binding site and colchicine binding site”, please check latest updates of the binding sites on Tubulin, there are more than four sites, for instance: pnas.org/cgi/doi/10.1073/pnas.1408124111.
  • polyphenol methyl ether structure is often seen in tubulin inhibitors, but this can be isosteric replaced, e.g., Med. Chem. 2018, 61, 17, 7877–7891. The authors should point out this in the Introduction part.
  • Should have a Chemistry part to include information like: chemical reagent and solvent source, TLC, NMR instrument, chemical shift, MS instrument…
  • For compounds A,B, C, spectrum data should also be provided, if known, please provide reference. Also, what are the yields for these intermediate?
  • Detailed synthetic procedures for all the compounds should be provided.
  • In the molecular modeling part (both Figure 4A and 4B), color should be adjusted to make it easy to look.
  • Line 50: αβ-tubulin dipolymer should be dimer
  • I suggest the authors consider calling the derivatives 3,4-Dihydro-2(1H)-quinolinone sulfonamide instead of quinoline sulfonamide.

Author Response

Dear reviewer,

We are very grateful to you for pointing out the errors in our manuscript before sending to peer reviewers. Your comments have been carefully incorporated into the revised version of the manuscript, which have significantly promoted the improvement of this manuscript. In addition, we have made a point-by-point response to these comments along with the cover letter. We hope that the current revised manuscript will be suitable to be published in this Journal.

Meanwhile, thank you very much for your time and effort to process our manuscript.

Best regards,

Sincerely yours,

Prof. Guo-Hua Gong

Inner Mongolia Key Laboratory of Mongolian Medicine Pharmacology for Cardio-Cerebral Vascular System, Tongliao, Inner Mongolia, China

Reviewer 1:

In this submitted manuscript titled “Discovery of Novel Quinoline-sulfonamide Derivatives as New Tubulin Polymerization Inhibitors with Anti-Cancer Activity”, Ma et al reported the synthesis of a series of 3,4-Dihydro-2(1H)-quinolinone sulfonamide derivatives and evaluated their anti-proliferative activity in four tumor cell lines. The authors found one of the derivatives called D13 showing the most potent activity and having IC50s at low-uM range. The authors also tried proving these derivatives are targeting microtubule dynamics and therefore performed tubulin polymerization assay and molecular modeling with the best compound D13.

The chemical structures and their anti-proliferative activities in this manuscript are of great interest. It is nevertheless both the polymerization assay and molecular modeling are not satisfying. At this stage, I do not recommend its publication in Molecules and revised version is required:

Major points:

The authors employed 5-fluorouracil as a positive control in Table 1, it is nevertheless 5-fluorouracil has an unrelated pharmacological target to the synthesized compounds presented in this manuscript. Therefore, I think it is inappropriate and I would suggest the authors to switch to tubulin inhibitors as positive controls.

Answer: Thank you for your suggestion. We determined the inhibitory effect of tubulin polymerization inhibitor CA-4 on HeLa, A549, HCT116, HepG-2 tumor cells, and set CA-4 as a positive control. This part of the data has been supplemented in Table 1.

The tubulin polymerization result is not convincing to me that D13 is inhibiting tubulin polymerization, also the positive control CA-4 does not display apparent tubulin polymerization activity as it supposed to. Can the authors please redo this assay or switch to assay measuring O.D. values?

Answer: Thank you for your careful review. I think this is a very serious problem, so we re-measured the inhibitory activity of compounds D13 and CA-4 on tubulin and increased the concentration gradient range. The results show that compounds D13 and CA-4 have a significant inhibitory effect on tubulin and have a good concentration dependence. (Fig. 3)

Molecular modeling: It is impossible for the proton of OCH3 to form hydrogen-bonding interaction, please redo the modeling with the help from expert.

Answer: Thank you for your remind. However, please check carefully, our molecular docking results did not indicate that OCH3 has an hydrogen-bonding interaction on the target protein. We re-analyzed the molecular docking of the target compound and tubulin. The results are shown in the Fig. 3.

Minor points:

In line 48, the authors state “Four tubulin inhibitor binding sites have been identified on microtubules, including paclitaxel binding site, Laulimalide binding site, vinblastine binding site and colchicine binding site”, please check latest updates of the binding sites on Tubulin, there are more than four sites, for instance: pnas.org/cgi/doi/10.1073/pnas.1408124111.

polyphenol methyl ether structure is often seen in tubulin inhibitors, but this can be isosteric replaced, e.g., Med. Chem. 2018, 61, 17, 7877–7891. The authors should point out this in the Introduction part.

Answer: Thank you for your suggestion. We have corrected and added relevant content.

Should have a Chemistry part to include information like: chemical reagent and solvent source, TLC, NMR instrument, chemical shift, MS instrument…

Answer: Thank you for your suggestion. We have provided the detailed chemistry part.

For compounds A,B, C, spectrum data should also be provided, if known, please provide reference. Also, what are the yields for these intermediate?

Answer: Thank you for your advice. We have added this part in the chemistry part.

Detailed synthetic procedures for all the compounds should be provided.

Answer: Thank you for your advice. We have provided the detailed synthetic procedures for all the compounds.

In the molecular modeling part (both Figure 4A and 4B), color should be adjusted to make it easy to look.

Answer: Thank you for your advice. We have followed your suggestion and polished the color of the picture, hope you will be satisfied.

Line 50: αβ-tubulin dipolymer should be dimer

Answer: Thank you for your advice. We have carried out the correction according to your suggestion

I suggest the authors consider calling the derivatives 3,4-Dihydro-2(1H)-quinolinone sulfonamide instead of quinoline sulfonamide.

Answer: Thank you for your advice. I have accepted your suggestion and amended the nomination of the article to "Discovery of Novel 3,4-Dihydro-2(1H)-quinolinone sulfonamide Derivatives as New Tubulin Polymerization Inhibitors with Anti-Cancer Activity"

Reviewer 2 Report

This research work presented in the manuscript entitled „Discovery of Novel Quinoline-sulfonamide Derivatives as New Tubulin Polymerization Inhibitors with Anti-Cancer Activity“ describe the synthesis of novel quinoline-sulfonamide derivatives in order to explore their biological activity. The chemistry part is well established by preparing the unknown compounds and the authors have described the synthesis, but not in details to be satisfied for this journal. The most significant part of the presented research, besides biological activity, is the inhibition of tubuline polymerization. The study is interesting and timely and is appropriate to the subject matter of the journal regarding bioorganic and medicinal chemistry but requires major revision. 

My other notifications are listed below:

  • The whole manuscript should be corrected for grammatical errors.
  • Please remove the masses and the molar quantity from the part Chemistry; it is unnecessary because it can be find in the Experimental part.
  • If there was some optimization of reactions, it should be explained. Also, they should also mention something about spectroscopic characterization, for example how the NMR spectroscopy has helped in the confirmation of prepared compounds, which specific signals obtained etc. Additionally, they did not mention the yields and the efficacy of the conducted reactions.
  • All compounds should be also tested for cytotoxic activity on normal cells. This is usual procedure when the tumor cells are used.
  • The discussion regarding the docking of chosen compounds should be also extended with clearly highlighted importance of these results.
  • In Materials and Methosds should be added the experimental procedure for each compound, not only NMR data. This is usual when a description of newly synthesized compounds is given.
  • 13C NMR spectra should be recorded and added to the Experimental part because it is usual to use 1H as well as 13C NMR for structural characterization of newly synthesized compounds.
  • Conclusion is missing and should be added with the clearly featured importance of obtained results, the possible application of prepared compounds as well as their further biological evaluation.
  • Author should clearly indicate which compound has been chosen as a lead compound for further optimization to achieve better activity

Author Response

Dear reviewer,

We are very grateful to you for pointing out the errors in our manuscript before sending to peer reviewers. Your comments have been carefully incorporated into the revised version of the manuscript, which have significantly promoted the improvement of this manuscript. In addition, we have made a point-by-point response to these comments along with the cover letter. We hope that the current revised manuscript will be suitable to be published in this Journal.

Meanwhile, thank you very much for your time and effort to process our manuscript.

Best regards,

Sincerely yours,

Prof. Guo-Hua Gong

Inner Mongolia Key Laboratory of Mongolian Medicine Pharmacology for Cardio-Cerebral Vascular System, Tongliao, Inner Mongolia, China

Reviewer 2:

This research work presented in the manuscript entitled „Discovery of Novel Quinoline-sulfonamide Derivatives as New Tubulin Polymerization Inhibitors with Anti-Cancer Activity“ describe the synthesis of novel quinoline-sulfonamide derivatives in order to explore their biological activity. The chemistry part is well established by preparing the unknown compounds and the authors have described the synthesis, but not in details to be satisfied for this journal. The most significant part of the presented research, besides biological activity, is the inhibition of tubuline polymerization. The study is interesting and timely and is appropriate to the subject matter of the journal regarding bioorganic and medicinal chemistry but requires major revision. 

My other notifications are listed below:

The whole manuscript should be corrected for grammatical errors.

Answer: Thank you for your advice. We have commissioned a professional organization to polish the language of the manuscript.

Please remove the masses and the molar quantity from the part Chemistry; it is unnecessary because it can be find in the Experimental part.

Answer: Thank you for your advice. We have removed the masses and the molar quantity from the part Chemistry.

If there was some optimization of reactions, it should be explained. Also, they should also mention something about spectroscopic characterization, for example how the NMR spectroscopy has helped in the confirmation of prepared compounds, which specific signals obtained etc. Additionally, they did not mention the yields and the efficacy of the conducted reactions. 

Answer: Thank you for your advice. We have provided detailed synthesis steps, melting points, yields, and appearance status of the intermediates, and provided 1NMR representing the intermediates.

All compounds should be also tested for cytotoxic activity on normal cells. This is usual procedure when the tumor cells are used.

Answer: Thank you for your advice. We have added the toxicity of the target compound to normal cells. Unfortunately, our target compound has similar cytotoxicity to tumor cells and normal cells. Regarding the optimization of chemical reactions mentioned by the reviewers, this research is a common reaction. I think there is nothing worth discussing.

The discussion regarding the docking of chosen compounds should be also extended with clearly highlighted importance of these results.

Answer: Thank you for your advice. We have redone the molecular docking research.

Two compounds D13 and D5 with better activity and one compound D15 with lower activity were selected as ligands for molecular docking. Through research, it is found that the binding ability of these three small molecule ligands and tubulin receptor is positively correlated with anti-proliferative activity.This part of the result can be added to the manuscript.

In Materials and Methosds should be added the experimental procedure for each compound, not only NMR data. This is usual when a description of newly synthesized compounds is given.

Answer: Thank you for your advice. We have added the specific synthesis steps of the target compound in the chemical part of the material part

13C NMR spectra should be recorded and added to the Experimental part because it is usual to use 1H as well as 13C NMR for structural characterization of newly synthesized compounds.

Answer: Thank you for your advice. We have measured 13C NMR for the target compound D1-D15, and the pop data has been added in the synthesis section of the article. However, because the amount of compound D16 is too small, its 13C NMR data has not been measured yet. If this is necessary, we will make it up later.

Conclusion is missing and should be added with the clearly featured importance of obtained results, the possible application of prepared compounds as well as their further biological evaluation.

Answer: Thank you for your advice. We have added the conclusion part of the article

Author should clearly indicate which compound has been chosen as a lead compound for further optimization to achieve better activity

Answer: Thank you for your advice. We have made some supplements in the conclusion. I think compound D13 can be used as a lead compound to further optimize the structure.

Round 2

Reviewer 1 Report

In this submitted revision, the authors have partially improved the manuscript according to my comments, nevertheless, there are still some major and minor points need fixing. Please see my comments below:

Major points:

  • In the revision, the authors are suggested to employ tubulin inhibitor as a positive control, although the authors have done so, it is nevertheless they just simply add IC50s of CA-4 without repeating together with some of the synthesized compounds, there could be variation existing if compounds are tested in cells on different days. I think it is more propriate to repeat some of the most potent derivatives (fi not all) together with positive control CA-4 on the same day. The authors are not required to put the result in manuscript but to show the result in the revision letter.
  • In the Modeling: the authors showed that there is binding energy difference for molecules 13 (-53 kJ/mol), 5 (-37 kJ/mol), and 15 (-31 kJ/mol) therefore claimed this may contribute to their IC50 differences. Nonetheless, this is not convincing to me, as we can clearly see that IC50 of compound 5 (IC50 ~ 1-3 uM) is similar to compound 13 (IC50 ~ 1-3 uM) but they have dramatically different in binding energy whilst compound 5 (IC50 ~ 1-3 uM) is far more potent than compound 15 (IC50 ~100 uM) but having similar binding energies (-37 vs -31 kJ/mol).

Minor points:

  • In line 46: Again, there are more than four binding sites have been reported on tubulin, the authors should claim how many binding sites exist on tubulin and then list names of the sites, I suggest the authors refer to a latest and comprehensive review.
  • In line 56-57: “Interestingly, the tubulin inhibitor CA-4 and BPROL075 (Fig. 1) also contain this structure”, the author should specify the polyphenol methyl structure.
  • In Scheme 1: sulfonyl chloride intermediate A is mislabeled as B; also the authors should show the structure of intermediate B since it is mentioned in the text.
  • In the main text line 83-91, font and size should be consistent to other part of manuscript.
  • Again, should have a Chemistry part to include information like: chemical reagent and solvent source, TLC, NMR instrument, chemical shift, MS instrument… (the authors ignored this in the revision)
  • Line 157, should it be compound A or B?

Author Response

Dear reviewer,

We are very grateful to you for pointing out the errors in our manuscript before sending to peer reviewers. Your comments have been carefully incorporated into the revised version of the manuscript, which have significantly promoted the improvement of this manuscript. In addition, we have made a point-by-point response to these comments along with the cover letter. We hope that the current revised manuscript will be suitable to be published in this Journal.

Meanwhile, thank you very much for your time and effort to process our manuscript.

Best regards,

Sincerely yours,

Prof. Guo-Hua Gong

Inner Mongolia Key Laboratory of Mongolian Medicine Pharmacology for Cardio-Cerebral Vascular System, Tongliao, Inner Mongolia, China

Reviewer 1:

Major points:

In the revision, the authors are suggested to employ tubulin inhibitor as a positive control, although the authors have done so, it is nevertheless they just simply add IC50s of CA-4 without repeating together with some of the synthesized compounds, there could be variation existing if compounds are tested in cells on different days. I think it is more propriate to repeat some of the most potent derivatives (fi not all) together with positive control CA-4 on the same day. The authors are not required to put the result in manuscript but to show the result in the revision letter.

Answer: Thank you for your suggestion. According to your suggestion, we selected compounds D13 and D5 with better activity and the positive control CA-4 to measure the anti-proliferative activity of tumor cells at the same time. The results are shown in Table 1 below. Compared with the data in the manuscript, there is no significant difference, indicating that the data in the manuscript is reliable.

Table 1. Antiproliferative Activities of D1、D5 and CA-4 against Human Cancer Cell Linesa

Compd.

R

(μM) IC50b

HeLa

A549

HCT116

HepG-2

L02

D5

4-CH3

1.42±0.3

2.82±0.13

1.15±0.08

2.46±0.21

3.45±0.21

D13

4-OCH3

2.13±0.21

1.24±0.18

0.89±0.31

1.78±0.93

3.12±1.18

CA-4

-

0.03±0.003

0.11±0.025

0.04 ±0.023

0.04±0.020

0.19±0.040

In the Modeling: the authors showed that there is binding energy difference for molecules 13 (-53 kJ/mol), 5 (-37 kJ/mol), and 15 (-31 kJ/mol) therefore claimed this may contribute to their IC50 differences. Nonetheless, this is not convincing to me, as we can clearly see that IC50 of compound 5 (IC50 ~ 1-3 uM) is similar to compound 13 (IC50 ~ 1-3 uM) but they have dramatically different in binding energy whilst compound 5 (IC50 ~ 1-3 uM) is far more potent than compound 15 (IC50 ~100 uM) but having similar binding energies (-37 vs -31 kJ/mol).

 Answer: Thank you for your suggestion. I have deleted the relevant discussion

Minor points:

In line 46: Again, there are more than four binding sites have been reported on tubulin, the authors should claim how many binding sites exist on tubulin and then list names of the sites, I suggest the authors refer to a latest and comprehensive review.

Answer: Thank you for your remind. I have corrected my exposition and revised it in the original manuscript. The modifications are as follows: By 2016, there were already seven active binding sites on tubulin. Among them, five binding sites are located on the β subunit of tubulin, including: paclitaxel binding site, laulimalide binding site, vinblastine binding site, maytansine binding site and colchicine binding site. Meanwhile, two binding sites are located on the α subunit of tubulin, including: evipabulin binding site and pironetin binding site [10,13-15].

In line 56-57: “Interestingly, the tubulin inhibitor CA-4 and BPROL075 (Fig. 1) also contain this structure”, the author should specify the polyphenol methyl structure.

Answer: Thank you for your careful review. However, I have made it clear in my previous discussion: the polyphenol methyl ether structure in its molecule is one of the main pharmacophores that inhibit tubulin polymerization. Therefore, I do not think it is necessary to repeat the discussion here.

In Scheme 1: sulfonyl chloride intermediate A is mislabeled as B; also the authors should show the structure of intermediate B since it is mentioned in the text.

Answer: Thank you for your careful review. However, compound B was mislabeled as A in the Synthetic Steps section, I have corrected it. Not mislabeled in Scheme 1.

In the main text line 83-91, font and size should be consistent to other part of manuscript.

Answer: Thank you for your careful review. I have made corrections to the fonts in this part.

Again, should have a Chemistry part to include information like: chemical reagent and solvent source, TLC, NMR instrument, chemical shift, MS instrument… (the authors ignored this in the revision) Line 157, should it be compound A or B?

Answer: Thank you for your careful review. I have added the description of the chemistry part, Line 157, should  be compound B.

Reviewer 2 Report

Authors have answered on each of my comments and sucesfully revised the manuscript. 

Author Response

Dear reviewer,

Thank you very much for agreeing to accept our manuscript. Thank you again for your hard work and dedication to the manuscript.

Best regards,

Sincerely yours,

Prof. Guo-Hua Gong

Inner Mongolia Key Laboratory of Mongolian Medicine Pharmacology for Cardio-Cerebral Vascular System, Tongliao, Inner Mongolia, China